# A Kinetic Approach to Oxygen Radical Absorbance Capacity (ORAC): Restoring Order to the Antioxidant Activity of Hydroxycinnamic Acids and Fruit Juices

**DOI:** 10.3390/antiox13020222

**Published:** 2024-02-09

**Authors:** Umme Asma, Maria Letizia Bertotti, Simone Zamai, Marcellus Arnold, Riccardo Amorati, Matteo Scampicchio

**Affiliations:** 1Faculty of Agricultural, Environment and Food Sciences, Free University of Bozen-Bolzano, Piazza Università 1, 39100 Bolzano, Italy; uasma@unibz.it (U.A.); simone.zamai@student.unibz.it (S.Z.); 2Faculty of Engineering, Free University of Bozen-Bolzano, Piazza Università 1, 39100 Bolzano, Italy; marialetizia.bertotti@unibz.it; 3Department of Gastronomy Science and Functional Foods, Faculty of Food Science and Nutrition, Poznań University of Life Sciences, Wojska Polskiego 31, 60624 Poznań, Poland; marcellus.arnold@up.poznan.pl; 4Department of Chemistry “G. Ciamician”, University of Bologna, Via Gobetti 83, 40129 Bologna, Italy; riccardo.amorati@unibo.it

**Keywords:** antioxidant mechanism, fluorescein decay, hydroxycinnamic acids, antioxidant activity, peroxyl and alkoxyl radicals, structure–activity relationship

## Abstract

This study introduces a kinetic model that significantly improves the interpretation of the oxygen radical absorbance capacity (ORAC) assay. Our model accurately simulates and fits the bleaching kinetics of fluorescein in the presence of various antioxidants, achieving high correlation values (R^2^ > 0.99) with the experimental data. The fit to the experimental data is achieved by optimizing two rate constants, *k*_5_ and *k*_6_. The *k*_5_ value reflects the reactivity of antioxidants toward scavenging peroxyl radicals, whereas *k*_6_ measures the ability of antioxidants to regenerate oxidized fluorescein. These parameters (1) allow the detailed classification of cinnamic acids based on their structure–activity relationships, (2) provide insights into the interaction of alkoxyl radicals with fluorescein, and (3) account for the regeneration of fluorescein radicals by antioxidants. The application of the model to different antioxidants and fruit extracts reveals significant deviations from the results of traditional ORAC tests based on the area under the curve (AUC) approach. For example, lemon juice, rich in ‘fast’ antioxidants such as ascorbic acid, shows a high *k*_5_ value, in contrast to its low AUC values. This finding underscores the limitations of the AUC approach and highlights the advantages of our kinetic model in understanding antioxidative dynamics in food systems. This study presents a comprehensive, quantitative, mechanism-oriented approach to assessing antioxidant reactivity, demonstrating a significant improvement in ORAC assay applications.

## 1. Introduction

The oxygen radical absorbance capacity (ORAC) assay is widely used to measure the antioxidant capacity of biological samples [1,2,3,4,5]. Initially proposed by Cao et al. [6], the ORAC assay uses fluorescein as a fluorescent probe, which loses its fluorescence upon exposure to oxygen radicals generated by the free radical initiator AAPH (2,2′-azobis(2-methylpropionamidine) dihydrochloride). The presence of strong antioxidants inhibits fluorescein decay, resulting in fluorescence signals that extend for longer periods of time. This effect can be conveniently quantified using the area under the curve (AUC) index, which provides a practical way to identify the antioxidant capacity [7]. A primary advantage of the AUC index is its ability to combine the induction time with the quantity of free radicals reacting with antioxidants. In addition, it is applicable to antioxidants, even those without a pronounced lag phase [8]. However, the mere integration of the fluorescein signal to represent antioxidant properties has several limitations. First, this method prioritizes the quantity of reacting antioxidants, giving relatively less importance to their reactivity toward radicals [5]. In addition, the units derived from the time-integrated fluorescein signals do not correspond to established chemical or physical quantities. This discrepancy renders the interpretation of ORAC units somewhat arbitrary. Furthermore, the ORAC protocol is subject to variations across different laboratories, leading to inconsistent AUC values and complicating inter-laboratory comparisons [9]. Lastly, another critical issue is the presumption that higher AUC values invariably indicate stronger antioxidant properties. This correlation has not been scientifically validated [5]. This problem becomes particularly evident when assessing certain antioxidants, such as hydroxycainnamic acids. In these cases, the ORAC-based ranking often does not align with the well-known structural properties [10,11].

In response to these challenges, kinetic modeling has gained growing interest, as shown by recent studies [12,13,14,15,16,17]. This approach, which has evolved significantly in recent years, employs numerical techniques to simulate time-dependent changes in the concentrations of reactants and products. The method typically begins with the definition of a comprehensive reaction mechanism. This step is followed by the derivation of a system of ordinary differential equations (ODEs) using the mass action law. Each ODE represents the rate of consumption of a reactant or the formation of a product. The subsequent numerical integration of these ODEs leads to accurate simulations of the transient concentration trends of each chemical species. Notably, the rates of these changes are governed by specific kinetic parameters, such as rate constants. These constants are iteratively adjusted until the model’s simulated curves match the empirical data, thereby enhancing the accuracy of the model. Achieving precision in aligning simulated curves with empirical observations substantially increases our understanding of complex chemical and biological processes. 

This perspective is particularly relevant in antioxidant research. Traditional methods, such as DPPH, CUPRAC, FRAP, and ORAC assays [4,18], while effective in assessing biomolecular activity, often do not provide insights into the chemical mechanisms involved [19]. Kinetic modeling, in contrast, offers a deeper understanding of the reaction dynamics between antioxidants and radical species. This approach represents a significant advance in antioxidant research, making conventional assays suitable for the study of reaction mechanisms.

The limited use of kinetic modeling in studies such as ORAC assays has primarily been due to the complexity of the computations and the challenging interpretation of the results. However, these problems have been greatly alleviated by recent advances in computational power and the development of user-friendly software. Nowadays, technological progress has made such analyses more accessible, even to researchers with limited expertise in this area. User-friendly tools, including KinGUII, CAKE, the main package for R, and COPASI, have revolutionized kinetic modeling approaches. They have simplified complex tasks, such as solving differential equations and optimizing parameters [20]. Among these tools, COPASI stands out for its user-friendly interface, which facilitates simulation and curve fitting. This simplicity has expanded its accessibility to a broader spectrum of researchers, as demonstrated by studies across various fields [21], including lipid peroxidation reactions [22] and antioxidant activity measurements [23].

In this study, we employed kinetic modeling to investigate the antioxidant activity of cinnamic acids and fruit juices using the simulation and fitting of classical ORAC assay data. Our model integrates factors identified in previous studies, such as the formation of peroxyl and alkoxyl radicals during fluorescein bleaching [24], their differential reactivity [25], and the multifaceted reactivity of antioxidants. This includes their scavenging activity toward peroxyl radicals and their capacity to repair fluoresceinyl radicals [26,27]. Despite the inherent complexity of this model, software such as COPASI simplifies the simulation and data fitting processes, making it more manageable. 

The validity of our model was confirmed by comparing the simulated fluorescein bleaching curves with the experimental data. We also examined the rate constants in relation to the established structure–activity relationships of antioxidants. This approach not only provides new insights into the study of antioxidant reactivity but also quantifies antioxidants through their reaction rates and kinetic rate constants. Thus, our study addresses the limitations of traditional ORAC assays and responds to the need for more precise and robust evaluation methods [24,28,29].

Although primarily aimed at food and pharmaceutical research, the implications of this study are broad, extending to industries such as cosmetics, polymers, and textiles, i.e., those sectors where effective antioxidant selection is crucial. The quantitative kinetic analysis presented here sets the stage for an enhancement in the information obtained from classical ORAC assay research.

## 2. Materials and Methods

### 2.1. Chemicals and Reagents

All chemicals and reagents were of analytical grade. Fluorescein (FH) and 2,2′-azobis(2-amidinopropane) dihydrochloride (AAPH) were obtained from Sigma-Aldrich (St. Louis, MO, USA). Throughout the experiments, distilled water was used. A 75 mM phosphate buffer solution (pH 7.0) was developed by combining equal volumes of 75 mM NaH_2_PO_4_ and 75 mM Na_2_HPO_4_ and then diluting to 1 L with distilled water. The pH was adjusted to 7.0 using a 6.0 M NaOH solution.

### 2.2. Preparation of Stock and Working Solutions

Fluorescein (FH): a 2 mM concentrated FH solution was prepared by dissolving the appropriate amount of fluorescein in distilled water. A 200 µM FH stock solution was then obtained by diluting the concentrated solution with phosphate buffer. This solution remained stable for 1 month when stored at 4 °C. Working solutions of FH, ranging from 0.05 to 3.2 µM, were prepared by further dilution with phosphate buffer.

AAPH: a 200 mM stock solution of AAPH was prepared daily by dissolving the requisite amount in phosphate buffer and used immediately.

Antioxidants: stock solutions of antioxidants (5 mM) were prepared in methanol. These were subsequently diluted with phosphate buffer to achieve concentrations ranging from 2.5 to 20 µM.

### 2.3. Sample Preparation

Fruits used in this study included apple (cv. Grany Smith and cv. Golden Delicious), orange, pineapple, grape, lemon, mango, and pomegranate. All samples were purchased in 2022 from a local supermarket in Bolzano (Italy) and were immediately analyzed on the day of purchase. Two replicates of each sample were prepared according to the method of Ou et al. [30] by treating independent batches of fruits or vegetables purchased in different weeks. In brief, each sample consisted of approximately 500 g of fruits or vegetables. The fruits were rinsed with water, dried with a cotton cloth, and cut into small pieces. The juice was collected using a commercial extractor (Moulinex JU650, Beijing, China) and centrifuged at 10,000× *g* rpm for 15 min (ThermoFisher Scientific, SL 16 Centrifuge Series, Dreieich, Germany). The supernatant was filtered using Whatman^®^ (Maidstone, UK) qualitative filter paper (25 mm diameter, Grade 4) and stored in a plastic Eppendorf^®^ (Hamburg, Germany) tube at −80 °C.

### 2.4. Oxygen Radical Absorbance Capacity (ORAC) Assay Procedure

The ORAC assay was performed at a controlled temperature of 37 °C using a Tecan Infinite M Nano Plus fluorescence spectrophotometer equipped with a dual-mode plate reader. In a Costar^®^ (Washington, DC, USA) 96-well black opaque plate, 50 µL of the fluorescein (FH) working solution was pipetted into each well. This was followed by the addition of 50 µL of either Trolox standard solution or the fruit extract (100-fold diluted) under investigation. After thorough mixing, the plate was incubated at 37 °C for 30 min to allow thermal equilibration. Subsequently, 100 µL of a pre-equilibrated 2,2′-azobis(2-methylpropionamidine) dihydrochloride (AAPH) working solution was introduced into each well, resulting in a final assay volume of 200 µL per well. The fluorescence intensity was continuously recorded for 120 min at specific excitation and emission wavelengths of 485 and 520 nm. The highest reading from the AAPH-free wells was used as the reference point for fluorescence measurements.

### 2.5. Kinetic Modeling

The kinetic model developed for this study is provided in the form of a Systems Biology Markup Language (SBML) file, and in the form of a Copasi file, both accessable as Appendix A. In addition, a comprehensive tutorial detailing the step-by-step procedures for the replication of the simulation and fitting processes of the kinetic model is available as Appendix A. This model encompasses nine distinct reactions, as shown in Figure 1. Each reaction contributes uniquely to the overall mechanism, which is defined by specific kinetic parameters, as reported in Table 1 (Reactions (1)–(9)). The roles and kinetic values of these parameters are elaborated in subsequent sections.

#### 2.5.1. Reaction (1): Initiation Reaction

In the ORAC assay, the initiation reaction is triggered by the thermal decomposition of 2,2’-azobis(2-methylpropionamidine) dihydrochloride (AAPH). This process, conducted in an environment where oxygen is not rate limiting, leads to the formation of nitrogen gas (N_2_) and two carbon-centered alkyl radicals (R•). These radicals may either rapidly recombine in a termination reaction or react with oxygen, forming peroxyl radicals (ROO•). The rate of ROO• generation, denoted as *R*_i_, is a critical factor in this reaction. *R*_i_ can be estimated through numerical fitting using a reference antioxidant, such as Trolox, with known stoichiometry for the trapping of ROO• radicals. Considering the surplus of AAPH (0.1 M), *R*_i_ was assumed to be constant throughout the reaction. To ensure reproducibility, conditions such as the oxygen concentration, pH, and temperature were controlled using a 0.1 M phosphate buffer at pH 7.0 and a temperature-controlled reaction chamber at 37 °C.

#### 2.5.2. Reactions (2) and (3): Peroxyl Radical Termination and Alkoxyl Radical Formation

In the absence of antioxidants, peroxyl radicals (ROO•) generated from AAPH primarily undergo self-reaction, leading to the formation of an unstable tetroxide intermediate (ROOOOR). However, in the case of tertiary peroxyl radicals, such as those derived from AAPH, this tetroxide does not decompose via the conventional Russell mechanism to yield closed-shell products. Instead, it forms alkoxyl radicals (RO•) and oxygen (O_2_). These RO• radicals are initially confined within the solvent cage, where they predominantly recombine to form non-radical products. Only a minority escape into the solution to participate in further reactions [24,25]. This sequence is modeled as two distinct reactions in our study: the first leading to non-radical products (outlined as Reaction (2) in Table 1), and the second producing two RO• radicals (described in Reaction (3) in Table 1). The relative weights of these two reactions were determined using Trolox as a reference antioxidant, and the optimal values obtained by the iterative fitting routine using the COPASI software were subsequently kept constant for further analyses.

#### 2.5.3. Reaction (4): Fluorescein Bleaching

Fluorescein (FH), the target indicator of the ORAC assay, primarily undergoes bleaching upon reaction with alkoxyl radicals (see Reaction (4) in Table 1). Here, its reaction with ROO• was considered negligible, which was also confirmed by independent inhibition experiments performed in the presence of THF, a substrate that rapidly reacts with peroxyl radicals (see Appendix A). The measurement of O_2_ consumption during THF oxidation in the presence of fluorescein confirmed the negligible affinity of fluorescein for peroxyl radicals (see Appendix A).

#### 2.5.4. Reaction (5): Antioxidant Activity

Within the assay system, antioxidants (AH) can intervene by neutralizing peroxyl radicals through a proton-coupled electron transfer mechanism (as illustrated in Reaction (5) in Table 1). This mechanism may involve either direct hydrogen atom transfer or electron transfer followed by protonation/deprotonation of the reactants [31].

#### 2.5.5. Reaction (6): Repair Mechanism

The assay also encompasses a ‘repair mechanism’, where antioxidants directly interact with fluorescein radicals (see Reaction (6) in Table 1). This reaction is rapid [29] and is primarily governed by the equilibrium constant *K*_6_. This constant reflects the relative difference in reduction potential Δ*E*^0^, as elucidated by Apak [2] and Bisby [29]. Higher values of *K*_6_ denote potent electron-donating activity, characterized by lower reduction potential *E*^0^ (A•, H^+^/AH) and, thus, a greater ability to regenerate fluorescein [29,32].

#### 2.5.6. Reactions (7)–(9) Radical Termination

The termination phase of the ORAC assay is characterized by the reduction of radical species, culminating in the formation of non-radical end products. This phase includes the self-decay of alkoxyl radicals (as detailed in Reaction (7) in Table 1), the decay of radicals derived from antioxidants (as shown in Reaction (8)), and the decay of radicals originating from fluorescein (outlined in Reaction (9)). The rate constants for these processes are generally considered significant, typically being on the order of 1 × 10^8^ M^−1^s^−1^ for most radicals and 1 × 10^9^ M^−1^s^−1^ for the more reactive alkoxyl radicals. Such rate constants underscore the velocity with which these radical species are converted into stable end products.

### 2.6. Kinetic Model Assumptions

The kinetic model for the ORAC assay observed in Table 1 encompasses nine reactions. For its practical use, it requires some assumptions because some rate constants are not directly measurable. The first assumption relates to Reactions (2) and (7) (Table 1), which involve the rapid termination of peroxyl and alkoxyl radicals. Because of their high reactivity, the termination reaction of peroxyl radicals was attributed to a rate constant *k*_2_ of 1 × 10^6^ M^−1^s^−1^, while that of alkoxyl radicals was attributed to *k*_7_ of 1 × 10^9^ M^−1^s^−1^. 

Another key assumption concerns Reaction (4). This indicates that fluorescein bleaching is predominantly mediated by alkoxyl radicals [10]. Given the high reactivity of alkoxyl radicals even with inactive substrates [33], a minimum rate constant *k*_4_ of 1 × 10^7^ M^−1^s^−1^ was assumed. Higher values of this constant have no impact on the bleaching kinetics of fluorescein.

Finally, Reactions (8) and (9) (Table 1) represent the self-termination events involving fluorescein and antioxidant radicals. The rate constants *k*_8_ and *k*_9_ are assumed to be 1 × 10^8^ M^−1^s^−1^. This assumption aligns with the observed kinetics and does not notably affect the fluorescein bleaching curve.

### 2.7. Kinetic Model Tuning

Considering the assumptions established for Reactions (2), (4), and (7)–(9) as listed in Table 1, the kinetic model was tuned by optimizing the kinetic parameters for Reactions (1), (3), and (6). This fine tuning was performed using Trolox as the reference antioxidant. Trolox was chosen because of its well-established stoichiometry (*n* = 2), which indicates its ability to trap two peroxyl radicals, as shown in Equation (1).
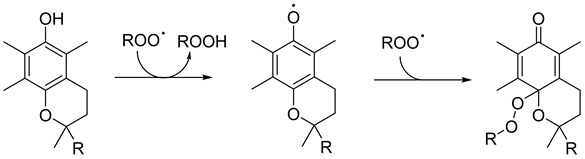
(1)


The use of Trolox is crucial in standardizing the kinetic model. This significance arises from the previously determined rate constant for Reaction (5) (*k*_5_ = 4 × 10^5^ M^−1^s^−1^) [31,34]. Consequently, Trolox provides a reliable reference standard for the calibration of the kinetic parameters of other reactions. Specifically, these adjustments include the initiation rate (*R*_i_) for Reaction (1), the rate constant *k*_3_ for Reaction (3), and the forward rate constant *k*_6_ for Reaction (6). The optimized values for these parameters are detailed in Table 2. Notably, the *k*_6_ value was determined by fixing the backward rate constant *k*_−6_ as 10^6^ M^−1^ s^−1^.

### 2.8. Simulation and Fitting of the ORAC Assay

The COPASI software (COmplex PAthway SImulator, version 4.8) was used for the simulation and fitting of the experimental fluorescein bleaching curves in the ORAC assay. On the basis of the reaction mechanism outlined in Table 1, COPASI generates a system of ordinary differential equations (ODE), each representing the rate of reactant formation or product consumption. The Livermore Solver for Ordinary Differential Equations (LSODA) algorithm efficiently solved these ODEs [35], by simulating temporal concentration changes. Optimal kinetic parameters, including rate constants and initial concentrations, were determined using the Stochastic Ranking Evolution Strategy (SRES) with the algorithm implemented in COPASI [36]. In brief, SRES iteratively generates a population of kinetic parameters, selecting those that best match the experimental data. This process iteratively refined the parameters until the simulated data aligned with the experimental results. A comprehensive tutorial on the use of COPASI is available as Appendix A.

### 2.9. Determination of the Antioxidant Activity (k_5_)

Antioxidant activity was expressed on the basis of the value of the rate constant *k*_5_. This was achieved using the iterative fitting program of the COPASI software, as outlined above. 

### 2.10. Determination of the Antioxidant Capacity (n)

Antioxidant capacity was measured based on the stoichiometric factor of the ORAC assay. This was achieved using the COPASI software by determining the optimal initial concentration of antioxidants. Thus, [AH]_0_ was considered a variable, and the software had to determine its optimal value to align the resulting simulated curve with the experimental data. Subsequently, the stoichiometry factor was calculated on the basis of the ratio between the optimal concentrations identified by COPASI (${\left[AH\right]}_{Copasi}$) to the actual concentration in the reaction mixture ${\left[AH\right]}_{Exp.}$, as outlined in Equation (2):(2)$$n=\frac{{\left[AH\right]}_{Copasi}}{{\left[AH\right]}_{Exp.}}$$

### 2.11. Determination of Fluorescein Repair Efficiency (K_6_)

The repair efficiency is expressed by the equilibrium constant (*K*_6_, see Reaction (6)). The *K*_6_ value was obtained by fixing the backward rate constant *k*_−6_ as 10^6^ M^−1^ s^−1^, while the forward rate constant *k*_6_ was iteratively optimized via fitting procedures. Subsequently, the equilibrium constant *K*_6_ was determined using Equation (3).
(3)$$K_{6}=\frac{k_{6}}{k_{-6}\;}$$

High values of *K*_6_ suggest that the equilibrium of Reaction (6) is shifted toward the products, demonstrating the ability of antioxidants to effectively reduce, or ‘repair’, the oxidized fluorescein. 

### 2.12. Computational Efficiency

Simulations and fittings were performed on a Lenovo ThinkPad (Intel(R) Core(TM) i5-8250U, 1.60–1.80 GHz, 8.00 GB RAM). The CPU time for the fitting of a 70–150 data point dataset was approximately 10 s, with approximately 24,000 function evaluations.

### 2.13. Statistical Analysis

Experiments were performed in triplicate. Data are presented as mean ± standard deviation (M ± SD). Statistical significance was determined using Student’s *t*-test or one-way analysis of variance (ANOVA), followed by Tukey’s post hoc test, with *p* < 0.05 as significant.

## 3. Results

### 3.1. Testing of the Kinetic Model

Figure 1 shows the simulation and fitting of the fluorescein bleaching curves using the kinetic model described in Table 1. This figure demonstrates a high correlation between the predicted and observed fluorescein bleaching curves in the presence of three antioxidants, specifically Trolox, sinapic acid, and coumaric acid. Notably, the experimental curves are consistent with those reported in previous studies by Dorta et al. [10,28], proving the reproducibility of the findings. The simulated curves fit the experimental data, with R^2^ values greater than 0.99. This accurate fitting was achieved by optimizing three key parameters, as reported in Table 3: the rate constants *k*_5_ and *k*_6_, which are crucial in expressing the reactivity of the antioxidants, and the initial concentration of antioxidants, which indicates the stoichiometry (Equation (2)). These parameters were estimated using the iterative fitting procedure, with uncertainties always below 5%, whereas experimental replicates exhibited uncertainty of less than 15%. The significance of these three parameters in the context of antioxidant characterization is discussed in the following sections.

### 3.2. Antioxidant Activity

Antioxidant reactivity, as indicated by the kinetic parameter *k*_5_, reflects the rate at which antioxidants neutralize free radicals. Trolox, a water-soluble analog of tocopherol, shows the highest reactivity (*k*_5_ = 4.0 × 10^5^ M^−1^s^−1^), a finiding that aligns with previous studies [34]. This high reactivity is due to its electron-dense chromanol ring, which facilitates hydrogen transfer and the resonance stabilization of the resultant radical [34,37]. Sinapic acid also shows considerable reactivity, with a *k*_5_ value of (60 ± 2) × 10^3^ M^−1^s^−1^. This can be attributed to its two *ortho*-methoxyl groups, which enhance the phenoxyl radical stabilization [38,39], albeit less than Trolox (*p* < 0.05). Ferulic acid (*k*_5_ = (30 ± 2) × 10^3^ M^−1^s^−1^), with one methoxyl group, and p-coumaric acid (*k*_5_ = (20 ± 2) × 10^3^ M^−1^s^−1^), with no methoxyl groups, exhibit lower reactivity [40]. Notably, the reactivity of sinapic acid is nearly three times that of p-coumaric acid, highlighting the role of methoxyl esters in enhancing antioxidant effectiveness [41].

Caffeic acid (*k*_5_ = (58 ± 2) × 10^3^ M^−1^s^−1^), with two hydroxyl groups and a conjugated double bond, and chlorogenic acid (*k*_5_ = (56 ± 2) × 10^3^ M^−1^s^−1^), similar to caffeic acid but with an additional quinic acid ester, also show substantial radical scavenging, although to a slightly lesser extent than sinapic acid.

Overall, the reactivity of these compounds, as quantified by *k*_5_, is consistent with previous studies on the AAPH-initiated peroxidation of human LDL [42] and with the nature of their aromatic substituents [39]. Electron-donating groups, such as hydroxyl and methoxy groups, increase the reactivity toward free radicals. Consequently, sinapic and caffeic acids, which have more effective electron-donating groups, show higher antioxidant activity compared to ferulic and p-coumaric acids. 

Interestingly, the kinetic model reveals an order of antioxidant reactivity that contrasts the results derived from the AUC approach. While the AUC index suggests p-coumaric acid as the most potent, followed by ferulic, sinapic, and Trolox, the kinetic analysis indicates otherwise, aligning with the electron density theory. This discrepancy calls for a cautious interpretation of antioxidant evaluations based solely on AUC values, as highlighted by the contrasting results observed here.

### 3.3. Repair Mechanism

In the ORAC assay, antioxidants exhibit additional functionality. They not only inhibit fluorescein bleaching by reacting with peroxyl radicals but also regenerate oxidized fluorescein, as detailed by Bisby [29] and Alarcon [43]. The capacity of antioxidants to restore fluorescein can be measured by the equilibrium constant *K*_6_, as shown in Reaction (6) in Table 1. Among the tested antioxidants, Trolox showed the highest repair capability for fluorescein. This ability is linked to its low redox potential (*E*_p,a_ = +80 mV) [26], which is significantly lower (more reducing) than that of fluorescein (*E*_p,a_ = +750 mV) [44]. The equilibrium constant *K*_6_ ranks the monophenolic cinnamic acids as sinapic acid > ferulic acid > p-coumaric acid. This ranking aligns with the findings of the DPPH assay [41,45]. Sinapic acid, with a lower reduction potential (*E*_p,a_ = +188 mV), exhibited the greatest regenerative potential, exceeding ferulic acid (*E*_p,a_ = +335 mV) and p-coumaric acid (*E*_p,a_ = +737 mV) [46]. Interestingly, the anodic peak of p-coumaric acid is lower than that of fluorescein (*E*_p,a_ = +750 mV) [44], suggesting its weak yet notable effectiveness in reducing oxidized fluorescein, despite its limited radical scavenging ability.

The ability of p-coumaric acid to repair oxidized fluorescein—combined with its weak radical scavenging activity toward peroxyl radicals—explains the unusually prolonged fluorescein bleaching curve observed in Figure 1. As p-coumaric acid is a weak peroxyl radical scavenger, the formation of alkoxyl radicals RO• via Reaction (3) is feasible. The formation of RO• explains the observed decline in the fluorescein signal prior to the induction time. However, the generation of RO• from ROO• decay is relatively inefficient, competing with the fast termination reactions involving alkoxyl radicals, resulting in the slower disappearance of p-coumaric acid compared to more reactive antioxidants. Overall, the prolonged bleaching curve observed in Figure 1 can be explained with the persistent presence of p-coumaric acid, its higher concentration than FH, and its capacity to regenerate oxidized fluorescein. Consequently, the high AUC values for p-coumaric acid, the least effective radical scavenger among the cinnamic acid series, can be attributed to its repair mechanism rather than its its radical scavenging activity.

In contrast, antioxidants such as Trolox, sinapic acid, ferulic acid, caffeic acid, and chlorogenic acid show shorter bleaching curves because of their stronger peroxyl radical scavenging activity. In their reduced states, these antioxidants effectively prevent alkoxyl radical formation by promptly reacting with peroxyl radicals immediately upon their generation through AAPH thermolysis. Therefore, their persistence in solution is proportionally shorter as a function of their reactivity with peroxyl radicals. Further variations in the reactivity between these antioxidants can also be attributed to their differing redox potentials, which influence their ability to restore the oxidized fluorescein, and thus affect the slope of FH bleaching prior to the induction time.

These results underscore the intricate mechanism of the ORAC assay. The observed fluorescein bleaching is not solely due to peroxyl radical scavenging by antioxidants but also due to their capacity to regenerate fluorescein in the presence of alkoxyl radicals.

### 3.4. Stoichiometry Factor

The stoichiometry factor, *n*, is a metric used to estimate antioxidant capacity. This is defined as the number of moles of reactive species that are scavenged by one mole of an antioxidant. As detailed in Table 3, these factors vary among different types of cinnamic acids. Monophenol cinnamic acids exhibit stoichiometry within 1.0 and 2.0. Notably, *ortho*-diphenolic cinnamic acids, such as caffeic acid and its methyl ester, exhibit stoichiometry of 4.0. This difference suggests that ortho-diphenols (catechols) have the ability to generate new active phenolic species, reasonably by the nucleophilic attack of the solvent on the orthoquinone, which is formed after donating two hydrogen atoms [47]. Caution is advised when interpreting these values because the experimental conditions, especially the reactant concentrations, can significantly influence them. For example, the results in Table 3 suggest that p-coumaric acid has a higher antioxidant capacity than sinapic acid and is comparable to Trolox, both exhibiting stoichiometry factors of 2. While stoichiometry reflects the molar ratio of antioxidants reacting with free radicals, it does not necessarily indicate overall antioxidant efficacy. For instance, p-coumaric acid and Trolox, despite similar stoichiometry values, differ markedly in their radical scavenging rates, as indicated by their respective *k*_5_ values in Table 3. Therefore, it is more informative to evaluate antioxidants on the basis of their reaction rates with radicals, rather than solely on capacity. Antioxidants with faster reaction rates are often more effective in radical scavenging, even if present in larger quantities. This distinction underscores the importance of assessing antioxidants based on the kinetic parameters of their reaction with free radicals.

### 3.5. Sensitivity Analysis

The capacity of our kinetic model to predict and fit the experimental decay of fluorescein bleaching has been rigorously tested under various experimental conditions. Figure 2A,B show the model fitting to the experimental data across different initial concentrations of fluorescein and Trolox. Higher fluorescein concentrations resulted in increased initial fluorescence signals, whereas increased Trolox concentrations extended the induction time of bleaching. In both cases, the model consistently demonstrated excellent fitting, as evidenced by the R^2^ values above 0.99, highlighting its robustness.

Figure 2C,D demonstrate the impact of *k*_5_ and *k*_6_ on antioxidant reactivity. Figure 2C shows that lower *k*_5_ values correlate with prolonged induction times, signifying a decreased rate of peroxyl radical scavenging. Conversely, Figure 2D reveals that higher *k*_6_ values result in distinct induction times and a ‘squared’ curve profile, indicative of an increased capacity of antioxidants to regenerate oxidized fluorescein. These findings are consistent with prior discussions, especially concerning the contrasting behaviors of Trolox and p-coumaric acid.

### 3.6. Application to Foods 

Extending the application of the kinetic model, we explored its performance in predicting antioxidant behavior in real food systems, such as fruit juices. The model was tailored to quantify the bleaching kinetics of fluorescein in diverse fruit extracts by determining the optimal *k*_5_ and *k*_5_ (or *K*_6_) values and the apparent initial antioxidant concentration ([AH]_Copasi_) through iterative fitting. As the concentration of the antioxidants in the samples is not precisely known, the stoichiometric factor cannot be determined. Thus, the [AH]_Copasi_ represents the product of [AH]_exp_ and *n* (see Equation (3)). Remarkably, Figure 3 reveals an order of antioxidant reactivity, based on *k*_5_, that significantly differs from the classical ORAC assay results: lemon ~ mango > orange ~ grape ~ pomegranate > pineapple > apple (Tukey HSD test, *p* < 0.05). The values of kinetic parameters *k*_5_ and *K_6_* contrast sharply with the AUC values from the classical ORAC assay, as indicated by the R^2^ value lower than 0.50, suggesting weak correlations.

A pertinent example illustrating the efficacy of the kinetic-based ORAC assay is the comparative analysis of lemon and apple juices. In traditional ORAC assays, which employ Area Under the Curve (AUC) values, apple juice exhibits notably high AUC values, suggesting superior antioxidant properties compared to lemon juice, characterized by comparatively lower AUC values. This inference, however, is incongruent with established culinary practices, wherein lemon is predominantly utilized for its antioxidative effectiveness in food preservation, rather than apple. Our kinetic model corroborates this empirical observation. It demonstrates that lemon juice, despite its lower rank in AUC-based methods, exhibits the highest reactivity, primarily attributable to its elevated *k*_5_ value. This divergence can be ascribed to the presence of rapid-acting antioxidants, such as ascorbic acid, in lemon juice, which swiftly neutralize radicals, thereby yielding lower AUC values. In contrast, apple juice contains antioxidants that show slower reactivity relative to those in lemon juice. This finding aligns with conventional gastronomic practices, affirming the validity of our kinetic-based approach in accurately delineating the antioxidant reactivity of food extracts [48]

Regarding the capacity of fruit extracts to regenerate oxidized fluorescein, most samples exhibited equilibrium constant *K*_6_ values above 1. This indicates the presence of antioxidants with lower redox potential than fluorescein. Mango, orange, and lemon juices notably recorded the highest *K*_6_ values, underscoring the presence of potent reducing agents.

These results demonstrate the potential of the kinetic-based approach as a versatile tool for antioxidant activity assessment in diverse food products. This kinetic approach not only accurately measures antioxidant reactivity but also provides deeper insights into antioxidative dynamics within real food systems.

## 4. Conclusions

Although the ORAC assay is an indispensable high-throughput method of assessing radical scavenging ability, its reliance on the Area Under the Curve (AUC) method can result in ambiguous interpretations. This study addresses these ambiguities by introducing and validating a kinetic model, thereby providing a more accurate representation of antioxidant reactivity in ORAC experiments. The proposed kinetic model allows the detection of the rate of the reaction of antioxidants with radicals. In addition, the kinetic model considers the crucial role of alkoxyl radicals in the bleaching of fluorescein and the dual role of antioxidants in scavenging peroxyl radicals and restoring oxidized fluorescein. 

Applying this model to individual antioxidants and a series of fruit extracts, we successfully determined absolute inhibition rate constants, providing a more objective, verifiable, and transferable measure of antioxidant activity than traditional AUC indices. This advancement enriches the application of the ORAC assay by moving from the conventional AUC method to a more nuanced kinetic approach. By using kinetic constants based on a precise reaction mechanism, our study not only improves the interpretation of ORAC assay results but also strengthens its reliability as an antioxidant assessment tool, paving the way for more accurate and comprehensive antioxidant research.

Future studies employing larger sample sizes would be invaluable in providing more definitive conclusions regarding the differences in antioxidant reactivity among various fruits, further enhancing the understanding and application of these findings in the field of food science and technology.

## Data Availability

The data that support the findings of this study are available on request from the corresponding author.

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
