# Peer review of "A Kinetic Approach to Oxygen Radical Absorbance Capacity (ORAC): Restoring Order to the Antioxidant Activity of Hydroxycinnamic Acids and Fruit Juices"

_antioxidants, 2024, doi:10.3390/antiox13020222_

Round 1
Reviewer 1 Report
Comments and Suggestions for Authors
The analysis of the antioxidant activity of various compounds is very interesting. An extensive analysis of the reactions occurring in the systems was carried out. This allows for a deeper understanding of the antioxidant properties of various compounds and food systems.
However, the article was missing some information:
Provide the kinetic equation on the basis of which the curves in Fig. 2 C and D were determined.
The range of independent variables for which the model can be used should also be provided. Does the model take into account temperature variability or can it only be used for reactions carried out at 37oC? Does antioxidant concentration influence the model, or is the model tailored to a specific antioxidant concentration? Is it possible to assess the impact of other substances found in food products on antioxidant activity?
Detailed comments
Line286: Missing variable description in equation 3
Line 464 -70 are probably template elements - please remove
Numbering error in equations 2 and 3
in the title of tab 3 there is: . The stoichiometry n is determined from eq. 4. There is no equation 4 in the text
Author Response
We greatly thank the reviewer comments. We have answered pointwise as it follows:
Q1. Provide the kinetic equation on the basis of which the curves in Fig. 2 C and D were determined.
Q1. In regard to the first comment "Provide the kinetic equation on the basis of which the curves in Fig. 2 C and D were determined.", the kinetic equation used to draw the curves in Fig. 2 C and D consists on the system of equations reported in Table 1. We added in the caption of the Figure 2 the kinetic parameters used to replicate the simulation. Also, in the Supplementary Information, it is present a tutorial that explain step by step how to replicate the results using the free software Copasi. Finally, we have included as Supporting information the file with the kinetic model in a XML format, which can be uploaded directly in Copasi or in other proprietary kinetic software.
Q2. (A) The range of independent variables for which the model can be used should also be provided. (B) Does the model take into account temperature variability or can it only be used for reactions carried out at 37oC? (C) Does antioxidant concentration influence the model, or is the model tailored to a specific antioxidant concentration? (D) Is it possible to assess the impact of other substances found in food products on antioxidant activity?
Q2. (A) We agree with the Reviewer for not having been enough clear in this important point. The range of independent variables for which the kinetic model was optimized are three: (1) the initial concentration of antioxidants, (2) the rate constant k5, and (3) the equilibrium constant K6. The other variables of the model were optimized in the previous experiment using Trolox as reference antioxidant. We have modified Table 1, defining in the column “Estimation” those variables that were tuned with experiments using Trolox, those who had a fixed value, and those who were considered a “Variable” to be tuned with the iterative fitting routine. We appreciate the Reviewer suggestion and we hope having improved the clarity of this important point.
(B) The model currently estimates absolute rate constants at 37°C, as this temperature is conventionally employed in the ORAC assay procedure. The primary objective of this manuscript was to propose a kinetic approach to the classical ORAC assay; therefore, we maintained experimental conditions, including temperature, consistent with the original protocol. Nonetheless, we concur with the reviewer that the kinetic approach has a general applicability and acknowledge that exploring its efficacy across different temperatures would be a valuable direction for future studies.
(C) The influence of antioxidant concentration on the derived rate constant in the kinetic model is limited. This is attributed to the fundamental principle that rate constants are, by definition, independent of reactant concentrations. The reviewer's query is pertinent, as excessively high initial concentrations of an antioxidant could indeed prolong the experiment's duration, potentially leading to uncontrolled and undesirable side reactions. To mitigate this, we capped the antioxidant concentration at a maximum of 20 µM in our experiments, a detail explicitly mentioned in the experimental section for clarity.
(D) The reviewer raises a significant point regarding the influence of other substances present in food products on antioxidant activity. However, the scope of our work was centered on refining the expression of ORAC assay results, rather than addressing its intrinsic limitations. The impact of other food constituents could certainly be explored, provided these constituents are soluble in the solvent used in the classical ORAC assay. Our study does not alter the experimental protocol of the ORAC assay; rather, it proposes a novel method of data analysis, transitioning from the area under curve approach to a kinetic-based approach
Q3. Line286: Missing variable description in equation 3
Q3. The rate constants in equation 3 have been better described in the text.
Q4. Line 464 -70 are probably template elements - please remove
Q4. We removed those template elements.
Q5. Numbering error in equations 2 and 3
Q5. We checked and revised the numbering of the equations.
Q6. in the title of tab 3 there is: . The stoichiometry n is determined from eq. 4. There is no equation 4 in the text.
Q6. We revised the numbering of the equations.
Reviewer 2 Report
Comments and Suggestions for Authors
1ºIn the validation of the "kinetic model" (lines 300-312) I do not understand what Table 3 contributes, since in Figure 1 what is reflected are the bleaching curves of fluorescein in the presence of three antioxidants: Trolox, Sinapic acid and Coumaric acid. However, Table 3 shows the kinetics of other antioxidant compounds (Ferulic acid, Caffeic acid and Chlorogenic acid) that have not been taken into account in Figure 1.
2ºI do not understand the reason WHY TABLE 3 APPEARS IN THE SECTION “VALIDATION OF THE KINETIC MODEL”. In my opinion, this table should appear after the "Stoichiometry factor" section (lines 396-415). Another option would be to change the order and put the "Stoichiometry factor" section after the "validation of the kinetic model" section.
3ºOn the other hand, the authors establish (lines 305-312) that the rate constants k5 and K6, as well as the initial concentration of antioxidants are key parameters when optimizing the fluorescein bleaching curves. However, in the Table 3, what is taken into account are only the constants K5 and K6. Can the authors explain to me how the initial concentration can influence this kinetic model, if this variable is precisely what we are seeking to quantify? In general, it seems to me difficult to follow reading the manuscript.
4º On line 449, the Figure is not numbered correctly. It should be numbered as Figure 3, instead of Figure 2.
5º In the "Sample Preparation" section, the authors include apples among the fruits analyzed. However, in Figure 3 they distinguish between the Grany apple and the Golden apple. These apple varieties should appear in this section. It would also be interesting if the authors collected the number of samples analyzed of each of the fruits and the conditions in which they were acquired and stored since these factors can affect the antioxidant composition.
6º As the authors say, assuming an average stoichimetry factor (n) of 2 is oversimplifying the diversity of antioxidant behaviors in food matrices. Therefore, I consider the conclusions reached by the authors to be very risky. In my opinion, opinion, the application of the kinetic model (K5) versus ORAC should be studied food by food, in greater depth and with a sufficiently high number of samples to be able to reach statistically significant results (p value). Precisely what is missing in the manuscript , in the application section to fruits, an adequate statistical treatment that allows me to know if the kinetic model really has a greater capacity to assess the presence of antioxidants than the ORAC method, given the complexity of the food matrices and the very different composition of compounds. antioxidants according to the fruit analyzed.
7º And finally, I don't understand what the inclusion of 3.1 means. Subsection and 3.1.1. Subsubsection,within the manuscript
Author Response
We greatly thank the Reviewer 2 comments. We have answered point-wise to each reviewer's comment as it follows:
Q1. In the validation of the "kinetic model" (lines 300-312) I do not understand what Table 3 contributes, since in Figure 1 what is reflected are the bleaching curves of fluorescein in the presence of three antioxidants: Trolox, Sinapic acid and Coumaric acid. However, Table 3 shows the kinetics of other antioxidant compounds (Ferulic acid, Caffeic acid and Chlorogenic acid) that have not been taken into account in Figure 1.
A1. We thank the Reviewer for this comment. Table 3 reports the kinetics parameters required to simulate the curve of Figure 1. Also, Table 3 reports the kinetic parameters to simulate curves from other antioxidants (ferulic, caffeic acid and chlorogenic acid), which have not been reported in Figure 1. Overall, Table 3 extends the results shown in Figure 1.
Q2. I do not understand the reason WHY TABLE 3 APPEARS IN THE SECTION “VALIDATION OF THE KINETIC MODEL”. In my opinion, this table should appear after the "Stoichiometry factor" section (lines 396-415). Another option would be to change the order and put the "Stoichiometry factor" section after the "validation of the kinetic model" section.
A2. We appreciate the constructive comment of the Reviewer. Our intention was to place Table 3 in the section “VALIDATION OF THE KINETIC MODEL” because the kinetic parameters used to simulate and fit the curves in Figure 1 are reported here. We have also cited Table 3 at line 321 page 8:
“This accurate fitting was achieved by optimizing three key parameters, as reported in Table 3: the rate constants k5 and k6, […], and the initial concentration of antioxidants, which indicates the stoichiometry (Eq. (2)).”
Although the Reviewer’s comment is valuable, as the positioning of Table 3 is important, we believe that the current placement of Table 3 is optimal, as it appears where citation exists in the text. However, we are open to editorial suggestions for improvement.
Q3. On the other hand, the authors establish (lines 305-312) that the rate constants k5 and K6, as well as the initial concentration of antioxidants are key parameters when optimizing the fluorescein bleaching curves. However, in the Table 3, what is taken into account are only the constants K5 and K6. Can the authors explain to me how the initial concentration can influence this kinetic model, if this variable is precisely what we are seeking to quantify? In general, it seems to me difficult to follow reading the manuscript.
A3. The Reviewer’s comment demonstrates interest and deep understanding of our results and for this we thank. Also, this question gives us the opportunity to explain that Table 3 reports not only the rate constants k5 and the equilibrium constant K6, but also the stoichiometry factor n, which, indeed, takes into account the initial concentration of antioxidants. The stoichiometry factor is the ratio between the concentration found by Copasi software during the iterative fitting process, and the initial antioxidant concentration added in the reaction vial. This ratio provides information of how many radical molecules can be scavenged by one antioxidant molecule.
Q4. On line 449, the Figure is not numbered correctly. It should be numbered as Figure 3, instead of Figure 2.
A4. The numbering has been checked and updated.
Q5. In the "Sample Preparation" section, the authors include apples among the fruits analyzed. However, in Figure 3 they distinguish between the Grany apple and the Golden apple. These apple varieties should appear in this section. It would also be interesting if the authors collected the number of samples analyzed of each of the fruits and the conditions in which they were acquired and stored since these factors can affect the antioxidant composition.
A5. The names of the apple varieties were added, together with additional information on the storage.
Q6. As the authors say, assuming an average stoichimetry factor (n) of 2 is oversimplifying the diversity of antioxidant behaviors in food matrices. Therefore, I consider the conclusions reached by the authors to be very risky. In my opinion, opinion, the application of the kinetic model (K5) versus ORAC should be studied food by food, in greater depth and with a sufficiently high number of samples to be able to reach statistically significant results (p value). Precisely what is missing in the manuscript , in the application section to fruits, an adequate statistical treatment that allows me to know if the kinetic model really has a greater capacity to assess the presence of antioxidants than the ORAC method, given the complexity of the food matrices and the very different composition of compounds. antioxidants according to the fruit analyzed.
A6. We thank the Reviewer for this important comment. After careful re-examination of this section:
- we recognize that assuming the stoichiometry value equal to 2.0 for all the samples sounds too arbitrary. The concept of stoichiometry, when applied to fruits, has little chemical meaning. Therefore, we considered the initial concentration of antioxidants [AH]Copasi as a variable to be optimized by the kinetic software. We have added an explanation of this concept in the section “Application to foods”.
- Regarding the reviewer's observation on the absence of adequate statistical analysis to ascertain whether the kinetic model surpasses the ORAC method in assessing antioxidant presence, we have now incorporated a statistical examination. This analysis specifically tests for differences in antioxidant activity among various fruits. We have amended Figure 3 to include annotations indicating significant differences. However, it is important to emphasize that these results should be considered preliminary. The sample size in this study was primarily designed to evaluate the practical utility of the kinetic model, rather than to comprehensively infer variability among different fruits. For this reason, we added in the Conclusion section a sentence that suggests that future studies with larger sample sizes could provide more definitive conclusions about the differences in antioxidant reactivity among fruits.
- Furthermore, concerning the question “if the kinetic model really has a greater capacity to assess the presence of antioxidants than the ORAC method”, we can conclude that the results of the kinetic-based ORAC assay can characterize antioxidants more deeply than the classical ORAC assay. This is because the kinetic-based approach allows to express the rate at which antioxidants react with radicals, while the classical ORAC assay measures only the amount of antioxidants that react with free radicals.
- Finally, we acknowledge the importance of studying the behavior of antioxidants in diverse food matrices, as pointed out by the reviewer. This was exactly the purpose of this section, showing the utility of the approach, which can be applied not only to characterize individual antioxidants, but also to study the behavior of complex samples.
We thank the Reviewer because based on his/her suggestions, we believe having enhanced the methodological robustness of the study.
Q7. And finally, I don't understand what the inclusion of 3.1 means. Subsection and 3.1.1. Subsubsection,within the manuscript
A7. These parts were residues from the journal's template. Thanks for highlight it, we have promptly deleted them now.
Reviewer 3 Report
Comments and Suggestions for Authors
The paper looks very important from several points of view. First of all food scientists desperately need analytical tools for establishing antioxidant properties and potential of food products. Methods should be accurate and should not interfere with side effects. Secondly the paper seems to be the good example of what we expect from real scientific paper by means not only a description of some phenomena but also goes deep down the understanding them including process/reaction mechanisms. After careful reading of the proposed paper I cannot see any discrepancies or things that needs to be upgraded or changes. The only one think is strange section 3.1. and 3.1.1. that are probably the remains of journal template. For the paper is ready for publication as it is.
Author Response
Q1. […] The only one think is strange section 3.1. and 3.1.1. that are probably the remains of journal template. For the paper is ready for publication as it is.
A1. We thank Reviewer n.3 for encouraging comments. We have amended section 3.1. and 3.1.1. which were remaining parts from the journal’s template. Thanks
Round 2
Reviewer 2 Report
Comments and Suggestions for Authors
I thank the authors for taking my observations into account and including them in this new version.
It significantly improves the understanding and reading of the manuscript and provides it with greater scientific quality.